# Quality adjustment and analysis of human resource prices in China: Based on a hedonic price model

Lan Ma [1,2] *

1 School of Mathematics and Statistics, Suzhou University, Suzhou, Anhui, P. R. China, 2 High Quality Economic Development Research Center in Northern Anhui, Suzhou University, Suzhou, Anhui, P. R. China

* malan@ahszu.edu.cn

## Abstract

The competition in the world has shifted from natural resources and capital resources to human resources. The human resources have become the real power source of the economic growth. Firstly, the price of human resources in China is calculated. Secondly, in order to measure the human resources quality adjustment index, the indicators system is constructed. Third, the Hedonic method is used to calculate the human resources "pure price" of 31 provinces (autonomous regions and municipalities directly under the Central Government) in China. The "pure price" has no the factor of human resources quality. Lastly, comparing the price of human resources before and after quality adjustment. The results show that: first, the number of human resources in China increased continuously during 1995–2015 and decreased during 2016–2020. Second, the price of nominal human resources in China keeps rising. In 2020, the nominal price is 39,087 yuan per person which is 15.44 times as many as in 1995. Thirdly, after the quality adjustment, the price of human resources has decreased significantly. The multiple between the actual and nominal price of human resources is between 1.75 and 2.12. Fourthly, the province with high human resource quality adjustment index generally have high quality human resource level or quantity. Fifth, the top five provinces of actual human resource prices are Shanghai, Beijing, Guangdong, Tianjin, Zhejiang, the bottom five provinces are Guizhou, Yunnan, Henan, Xizang, Gansu. Finally, the paper puts forward some policy recommendations: Improving the data collection mechanism of human resources accounting to provide a basic guarantee for the accurate accounting of human resources. Improving the price of human resources in the central and western regions to attract the talents to transfer to the central and western regions. Enhancing the skills training of human resources to improve the quality of human resources in the western region.

## 1.Introduction

According to the "China Human Development Report 2019" released by the United Nations Development Program, China is one of the most rapidly advancing countries in the field of

---

**Citation:** Ma L (2024) Quality adjustment and analysis of human resource prices in China: Based on a hedonic price model. PLoS ONE 19(4): e0297352. https://doi.org/10.1371/journal.pone.0297352

**Data Availability Statement:** All relevant data are within the manuscript and its Supporting Information files.

**Funding:** 1. The Humanities and Social Science Research Project of Anhui Universities,

SK2021A0696 2. A Doctoral Research Foundation Project at Suzhou University 2021, 2021BSK015 3. The Online course of Anhui Province—National Economic Accounting, 2021xskc097.

**Competing interests:** The authors have declared that no competing interests exist.

human development. Based on the Human Development Index (HDI), China has become a "high-level human development country" and is one of the fastest-growing countries in the field of human development in the past 30 years. China's Fourteenth Five-Year Plan puts forward the strategy of "Strengthening the Nation with Talents". In order to raise the level of human resources in the country, the cultivation, introduction, use and motivation of talents should be emphasized. Human resources are put into the labor market as labor that can create social wealth and value. According to the Marx's theory of labor value, the labor can be regarded as a "special commodity" [1]. Prices would generate in the process of commodities exchange. Adam Smith said in The Wealth of Nations that the labor is a commodity just as other commodities. The labor has a market price and a natural price. The wages are labor's market price [2]. The price of a commodity is a monetary expression of its value. The labour force has the characteristic of value and correspondingly has the characteristic of price. The theoretical basis of human resource price research are Marx's labour value theory and Adam Smith's labour price theory. At the same time, the paper draws on the accounting content of the System of National Economic Accounting 2008 (SNA2008), the Social and Demographic Statistics System (SSDS) and China's System of National Accounts 2016 (CSNA2016).

Human resources have both quantitative and qualitative characteristics. The quantity of human resources is the total number of human resources owned by a region, while human resource quality reflects the type of quality and complexity of the population (employees) in a region. Currently, the economy has entered the stage of high-quality development, which presents higher requirements for the quality of human resources. Research on the quality of human resources has therefore received increasing attention. The early institution for the systematic study of the Population Quality Index was the US Overseas Committee. It proposed the Population Life Quality Index in 1977, which comprehensively reflects the level of population development and compares the population quality in different regions and at different times. Human resource quality is the core element for measuring human resources and the reflection of human resource ability. Scholars have increasingly paid attention to the quality of human resources. Li and Tang [3] constructed China's labour quality index by using the J-F lifetime income method and the current income difference method. Huang et al. [4] studied the impact of human resource quality on the income of employees in a company and demonstrated human resource quality by the proportion of the population at or above the college level. Yue et al. [5] noted that China needs to develop high-quality human resources. Compared with the international developed level, the proportion of talent with high academic degrees and a high employment structure must be further optimized, and the degree of matching employment needs to be improved.

Scholars generally focus on the human resources in enterprises. There are few studies on the human resources at the national level. Ahdar, Musyarif [6] stated that the media education plays an important role in human resource enhancement. Juliana Jaya P E [7] stated that through human resource quality, local culture, and product performance, the competitiveness and well-being could be improved in firms.

The aims of the research in this paper is as follows: First, exploring the status of the quantity and price of human resources in China. Second, establishing the human resource quality index system. Using the hedonic method to calculate the human resource quality adjustment index. Calculating the actual human resource prices. Finally, comparing and analyzing the difference between nominal and actual human resource prices.

The research methods in this paper include the descriptive statistical analysis, the Hedonic feature price method, and the principal component analysis.

The contribution of this paper lies in the following: (1) Making up the gap in human resource price adjustment.It is conducive to improving the analytical framework of human

resource price adjustment and quality analysis. (2) It is helpful for accelerating the improvement of the statistical practice of human resource accounting and expanding the content of population accounting. (3) This paper provides the basis for the government to formulate a reasonable policy of regional human resource allocation, which is an important impetus for the promotion of high-quality development of the economy and the implementation of the strategy of "Talents Strengthening the Nation".

## 2.Literature review

The literature review section will be presented in four parts: Quantity of human resources, Price of human resources, Quality of human resources, and the Method of adjusting the quality of human resources.

### 2.1. Quantity of human resources

The quantity of human resources is the quantity of the labour force in a region. Specifically, the quantity of human resources = the working-age population–the incapacitated population + the underage employment population + the elderly employment population. Limited by the availability of data, the current index reflecting the number of human resources is mainly the quantity of effective labor force. Combined with existing studies [8–10], it is reliable and operable to use the number of employees to reflect the quantity of human resources in a certain region.

### 2.2. Price of human resources

The literature generally uses the average wage as a measure of labour price. In foreign countries, Farr [11] first used the average wage of workers to analyse the price of the rural labour force and used the present value of residents' lifetime labour income as residents' human capital. Jeong [12], You et al. [13], Bobeica et al. [14] used residents' wages as labour prices to study regional human capital levels. In China, Lin [15], Li and Zhang [16] and Wang [17] all used the average wage level as the measurement standard of human resource prices. Hu [18] believed that the price of rural labour consists of two parts: one is the wage level of migrant workers, and the other is the wage level of rural employees. Yi and Yang [19] used the labour price per mu of rice and the daily wage of agricultural hired labour to reflect the price of rural labour. Shen and Zhang [20] used the total salary of employees in urban units to represent the labour price. Therefore, the average wage of labour is an effective indicator to measure the price of human resources.

### 2.3. Quality of human resources

Previous studies on human resource quality are generally based on different dimensions. According to the requirements of the Social and Demographic Statistics System (SSDS) issued by the United Nations, Wang [21] used the matrix method to explore the total social population, structure and changes, analyse the production process and worker skills, and make a comprehensive evaluation of the population quality. Zhao and Zhu [22] reflect the level of human capital through the quantity, structure and concentration degree of human capital investment. Hao [23] and Wang [24] reflect the quality of human capital by the level of education. Zhao [25] reflected the quality of human resources by using teachers' salaries, the ratio of teachers to students in colleges and universities, education expenditure per student and education rate of return.

## 2.4. Quality adjustment method

Existing studies generally adjust the quality of commodity prices. Griliches [26] used the characteristic price index method to adjust the quality of automobile prices, and an increasing number of studies have been conducted on price adjustment using this method. Dulberger [27] made quality adjustments of computer prices. Lei [28] improved the characteristic price model and adjusted the price index of big data of China's e-commerce platforms. Shi [29] compiled a mobile phone price index based on hedonic price theory. Xu [30] conducted quality adjustment of China's CPI in the context of digital economy. Currently, it is rare to adjust the quality of the human resource price, but it has a sufficient theoretical basis.

SNA2008 noted that the quality adjustment procedure is tailored to specific goods and mainly adopts a feature model method. The hedonic method is a typical feature model method and can be traced back to the 1930s. The main idea of this method is to regard commodity prices as a function of commodity characteristics. Different characteristics and qualities of commodities determine different prices of commodities.

Court [31] is the first to use the hedonic method to adjust commodity quality, focusing on the research object of automobile prices, which are regarded as a function of automobile weight, length, fuel consumption, engine energy efficiency and other factors to establish the automobile hedonic price index. Subsequently, the hedonic method has become more widely used in pricing agencies in various countries. Some European countries have also applied the hedonic method to CPI products such as computers and cars. In China, there are also an increasing number of studies on quality adjustment using the hedonic method. Chen and Hu [32] both used the hedonic method to study the compilation of price indices. Chen [33] adjusted the quality of online data using the hedonic price index method.

Through studying the existing literature, it is found that the existing studies on human resources have the following shortcomings: (1) There are more studies on the quantity of human resources, but there is a lack of the comparisons between regions. (2) There are fewer studies on the price and quality of human resources, and quality adjustment of human resources price is even rarer.

In this paper, the number of effective labor force is used to measure the quantity of human resources in conjunction with existing studies. The average wage of labor force is used to measure the price of human resources. The quality of human resources is measured in different dimensions by establishing an indicator system. Based on the macro data of Chinese human resources, this paper does the following works: (1) To analyse the changes in the quantity, price and quality of human resources in China. (2) To calculate the human resource quality adjustment index. (3) To survey the differences in human resource prices in 31 provinces (autonomous regions and municipalities) before and after quality adjustment.

## 3.Data description and human resource development status

### 3.1. Data description

The data used in this paper are obtained from the Statistical Yearbook of China and various provinces, the Statistical Yearbook of Chinese Population and Employment, and the EPS database. Given the limitations of the completeness of the data, the data interval in this paper is 1995–2020.

### 3.2. Current situation of human resource quantity

Combined with existing research, this paper uses the "number of employees" to reflect the amount of human resources in a certain area, which is reliable and operable. The number of

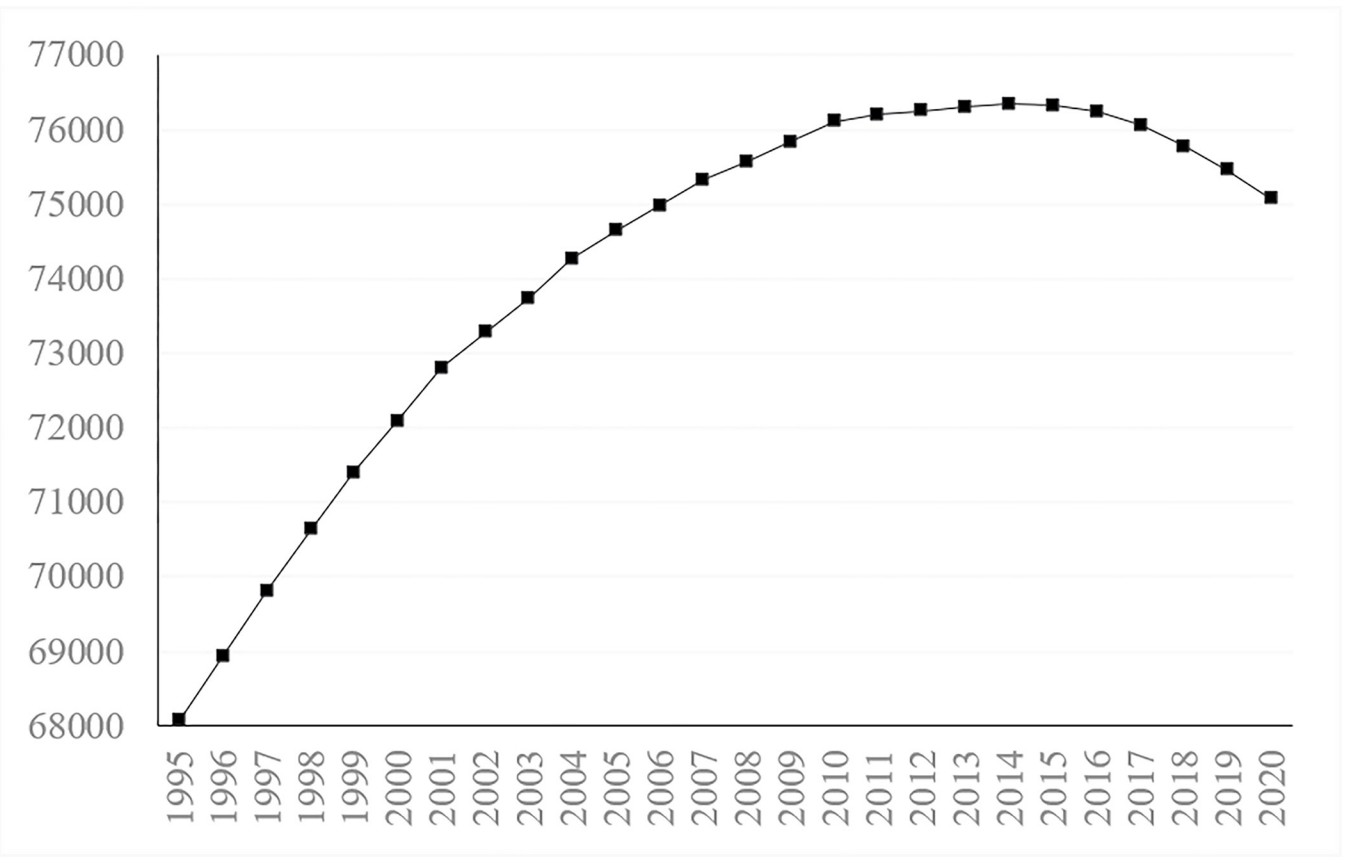

**Fig 1. Amount of HR in China from 1995 to 2020 (ten thousand people).**

employees is the amount of human resources that are put into production and creates value for society. Fig 1 shows the change in the number of human resources in China from 1995 to 2020.

Fig 1 shows that the number of employees in China from 1995 to 2020 showed an overall trend of first rising and then declining. The average growth rate of the number of human resources in China is 0.39%. In 1995, the number of employed persons was 680.65 million, which continued to rise to 763.2 million in 2015. The decline began in 2016, and the number of employees in 2020 was 750.64 million.

Fig 2 shows the average number of employees in the 31 provinces in China from 1995 to 2020 and is arranged in order from large to small. The line chart is the average growth rate of the number of employees in the corresponding province from 1995 to 2020.

Fig 2 shows that provinces with a high quantity of human resources are related to local population resources and economic development. The specific findings are as follows.

First, in terms of the absolute quantity of human resources in each province, Shandong, Henan, Guangdong, Sichuan and Jiangsu rank as the top five provinces with 59.46 million, 57.9 million, 53.5 million, 47.42 million and 46.18 million, respectively. The bottom five provinces are Tianjin, Hainan, Ningxia, Qinghai and Tibet, with 6.64 million, 4.34 million, 3.15 million, 2.98 million and 1.72 million employees, respectively.

Second, in terms of the average growth rate of the amount of human resources in each province, the five provinces with the fastest growth rates are Xinjiang, Guangdong, Beijing, Shanghai and Tibet, with growth rates of 2.82%, 2.77%, 2.26%, 2.22% and 2.09%, respectively.

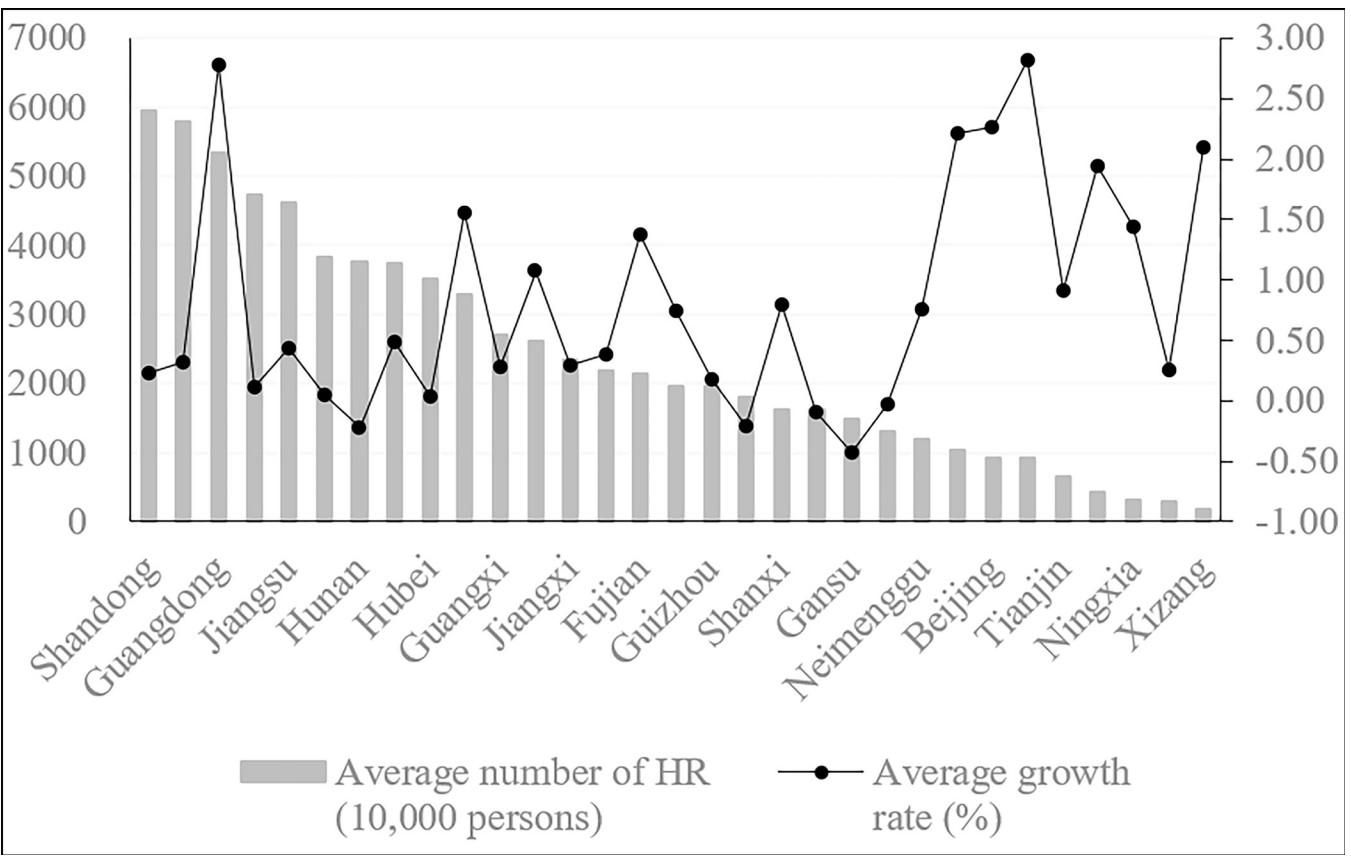

**Fig 2. Average amount of HR and average growth rates in China's 31 provinces.**

The five provinces with the slowest growth rates are Jilin, Chongqing, Heilongjiang, Hunan and Gansu, with growth rates of -0.03%, -0.09%, -0.21%, -0.22% and -0.43%, respectively.

Third, although the amount of human resources in Xinjiang, Xizang and Ningxia ranks low, their average growth rate is high. The western regions represented by these three provinces are increasingly attracting human resources and have achieved good results. Beijing, Shanghai and Tianjin, as municipalities directly under the central government, are not superior in the absolute number of employees compared with the provinces with large populations, but the growth rate of their employees is considerable. As a municipality directly under the central government, in Chongqing, the growth rate of employees is negative (-0.09%), which is related to its geographical location. Chongqing should continue to create more job opportunities and attract more employees. Hunan, Hubei, Anhui and Sichuan, as the dominant provinces for human resources, have a low average growth rate. These provinces need to pay attention to the coordinated development of human resources while developing their economies.

### 3.3. Current situation of human resource prices

According to Marx's labour value theory and Adam Smith's wage theory, human resources are valuable and have the same price attributes as commodities. The price of human resources actually refers to the compensation obtained by the human resources owner who transfers the right to use it; specifically, it is the labour remuneration of workers. This section estimates the

price of human resources in 31 Chinese provinces (autonomous regions and municipalities) with the help of residents' wages. Wages in urban and rural areas are two different statistical categories, so they need to be calculated separately. The basic idea of accounting is to adopt the per capita wage income of urban residents and the per capita wage income of rural residents to represent the labour price of urban and rural areas, respectively, and take the proportion of urban and rural employees as the weight for the labour price of the available area, which is the price of human resources studied in this paper.

**3.3.1. Calculation formula for human resource prices.** Human resource prices can be divided into urban human resource prices and rural human resource prices. The price of human resources in the labour market is the labour wage. The calculation formula is as follows:

$$\text{Urban per capita wage income} \times \text{Urban population} = \text{Total urban wage income} \quad (1)$$

$$\text{Price of urban human resources} = \frac{\text{Total urban wage income}}{\text{Number of urban employees}} \quad (2)$$

$$\text{Rural per capita wage income} \times \text{Rural population} = \text{Total rural wage income} \quad (3)$$

$$\text{Price of rural human resources} = \frac{\text{Total rural wage income}}{\text{Number of rural employees}} \quad (4)$$

$$\text{Price of human resources} = \frac{\text{Total urban wage income} + \text{Total rural wage income}}{\text{Number of urban employees} + \text{Number of rural employees}} \quad (5)$$

The price of urban human resources can be calculated by Formula (1) and Formula (2), and the price of rural human resources can be calculated by Formula (3) and Formula (4). The price of human resources in China can be obtained through the combination of Formula (5).

**3.3.2. Calculation results of human resource price.** According to Formula (1)–Formula (5), the human resource prices of the nation and 31 provinces (autonomous regions and municipalities) can be calculated from 1995 to 2020. The results are shown in Fig 3 and Table 1.

**(1) National Human Resource Price.**

Fig 3 shows the national and urban–rural human resource prices.

First, the price of human resources in China showed an overall upwards trend from 1995 to 2020. The price of human resources in urban areas was consistently higher than that in rural areas, and the absolute gap between them gradually widened from 5,589 yuan per person in 1995 to 39,087 yuan per person in 2020.

Second, the ratio of urban and rural human resource prices has been gradually reduced. The ratio decreased from 10.01 times in 1995 to 4.16 times in 2020. The average growth rate of urban human resource prices is 8.83%, while that of rural areas is 12.71%. Although the price of rural human resources is low, its growth rate is considerable.

Third, the price of human resources in 2013 fell briefly. On the one hand, due to the change in statistical calibre, the per capita net income of rural residents was counted before 2013 and the per capita disposable income of rural residents began to be counted after 2013, similar to urban residents. On the other hand, it is related to the economic development of that year. The growth rate of China's GDP in 2013 was 0.28 percentage points lower than that of the previous year, which led to a decline in the growth rate of wages.

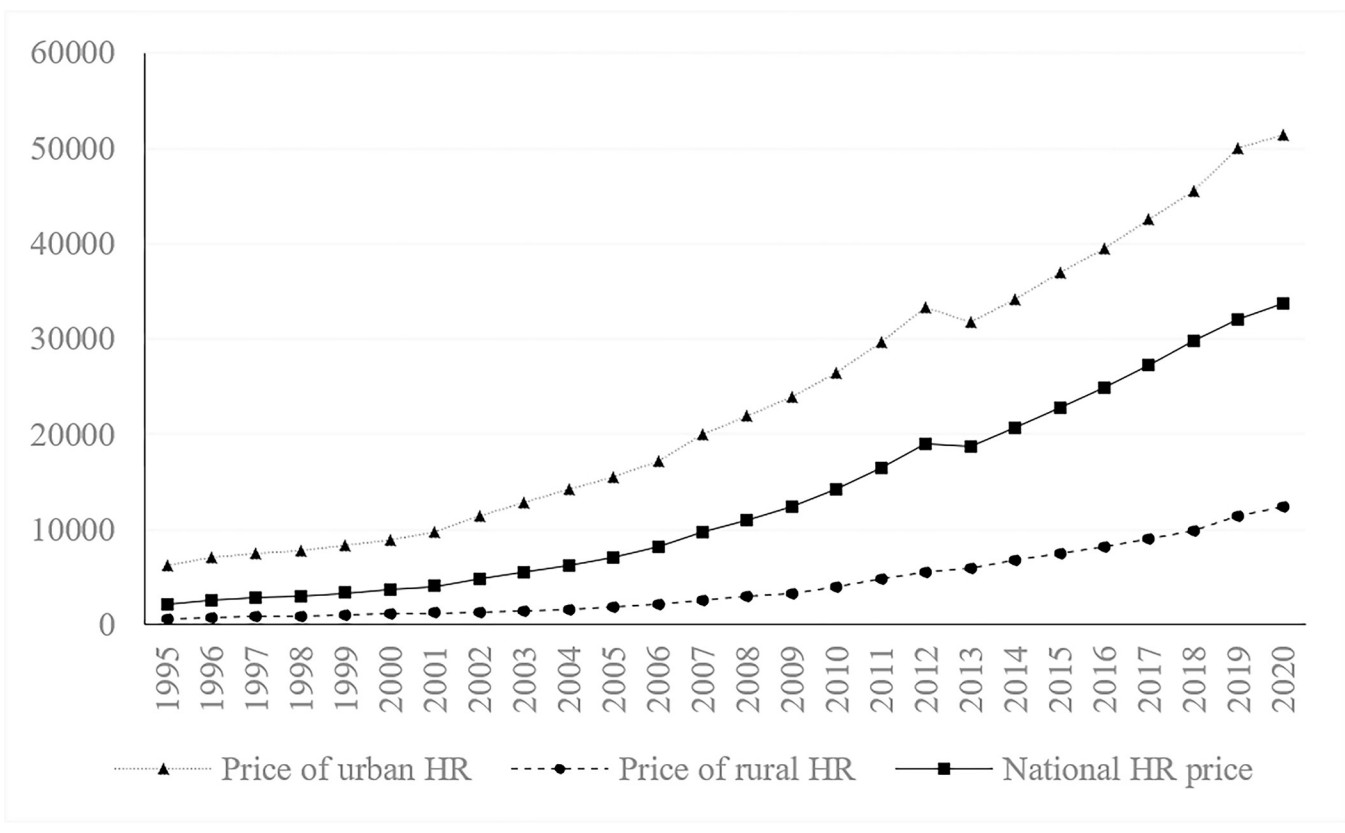

**Fig 3. National and suburban and rural HR prices from 1995 to 2020 (yuan/person).**

**(2) Human Resource Prices of Each Province.**

The human resource prices of 31 provinces (autonomous regions and municipalities) in China from 1995 to 2020 can be calculated by the same method. Due to space limitations, the results of some years are shown in Table 1 and sorted by the mean value of human resource prices from highest to lowest.

Table 1 shows the following.

First, the top five human resource prices all belong to the eastern region, while the bottom five include four provinces in the western region and one in the central region. The top five provinces in the average value of human resource prices are Shanghai, Beijing, Guangdong, Tianjin, and Jiangsu, and the human resource prices are 37,518 yuan, 34,490 yuan, 21,641 yuan, 18,594 yuan, and 18,258 yuan, respectively. The bottom five provinces are Guizhou, Henan, Gansu, Yunnan and Xizang, with human resource prices of 9,002 yuan, 8,724 yuan, 8,336 yuan, 8,244 yuan and 7,914 yuan, respectively.

Second, the top provinces in HR prices are ranked at the bottom or middle of the growth rates, such as Guangdong and Beijing. Henan, Gansu, Anhui and other provinces with low labour prices are growing faster. The five provinces with the fastest average growth rates of human resource prices are Hainan, Henan, Gansu, Anhui and Ningxia, with average annual growth rates of 15.44%, 15.03%, 14.64%, 14.09% and 13.98%, respectively. The five provinces with the slowest growth rates are Heilongjiang, Beijing, Xinjiang, Guizhou and Guangdong, with average annual growth rates of 9.88%, 9.82%, 9.11%, 8.99% and 8.56%, respectively. The provinces with lower human resource prices are gradually raising the corresponding prices to absorb more excellent talent.

Table 1. HR prices in 31 provinces of China (part of years) (yuan/person).

| Province | 1995 | 1996 | 2007 | 2008 | 2018 | 2019 | 2020 | Mean | Average growth rate(%) |
|---|---|---|---|---|---|---|---|---|---|
| Shanghai | 6278 | 7336 | 41613 | 48492 | 60406 | 69000 | 77594 | 37518 | 10.58 |
| Beijing | 7740 | 9272 | 27560 | 30471 | 65608 | 73062 | 80516 | 34490 | 9.82 |
| Guangdong | 7243 | 7037 | 18873 | 19842 | 44007 | 50230 | 56454 | 23403 | 8.56 |
| Tianjin | 3096 | 3640 | 16625 | 19707 | 43849 | 53422 | 62994 | 21641 | 12.81 |
| Jiangsu | 2761 | 3518 | 13542 | 15161 | 38455 | 44623 | 50790 | 18258 | 12.35 |
| Zhejiang | 4723 | 5401 | 14282 | 15398 | 28751 | 43230 | 57708 | 17715 | 10.53 |
| Fujian | 3475 | 4102 | 12226 | 13820 | 28024 | 39046 | 50069 | 15596 | 11.26 |
| Liaoning | 3074 | 3491 | 11544 | 13426 | 30558 | 35601 | 40643 | 14840 | 10.88 |
| Neimenggu | 2068 | 2264 | 11277 | 12663 | 29487 | 36202 | 42918 | 13969 | 12.9 |
| Shanxi | 2119 | 2322 | 10593 | 10793 | 28771 | 32287 | 35802 | 13503 | 11.97 |
| Shandong | 1737 | 2225 | 10107 | 11181 | 29037 | 36758 | 44480 | 13420 | 13.85 |
| Chongqing | 2001 | 2310 | 11450 | 13149 | 26869 | 33747 | 40626 | 13262 | 12.8 |
| Xijiang | 7740 | 9272 | 9754 | 10476 | 24343 | 30104 | 35866 | 12629 | 9.11 |
| Hebei | 2134 | 2486 | 8335 | 9145 | 27175 | 34686 | 42196 | 12314 | 12.68 |
| Jinlin | 1918 | 2841 | 10944 | 12003 | 23072 | 29753 | 36435 | 12242 | 12.5 |
| Ningxia | 1640 | 1755 | 7794 | 9468 | 26070 | 34659 | 43248 | 12045 | 13.98 |
| Jiangxi | 1706 | 1908 | 7962 | 8869 | 25934 | 34335 | 42735 | 11807 | 13.75 |
| Shanxi | 1865 | 2474 | 7337 | 8831 | 24599 | 29833 | 35067 | 11580 | 12.45 |
| Heilongjiang | 3489 | 3785 | 8580 | 9302 | 21350 | 29049 | 36748 | 11569 | 9.88 |
| Hainan | 1069 | 1768 | 8891 | 9915 | 23325 | 31030 | 38735 | 11569 | 15.44 |
| Hunan | 2151 | 2454 | 7370 | 8117 | 25380 | 32985 | 40590 | 11405 | 12.47 |
| Qinghai | 1694 | 2023 | 6686 | 7525 | 24208 | 32375 | 40541 | 10782 | 13.54 |
| Hubei | 1837 | 2000 | 7495 | 8292 | 22140 | 27519 | 32899 | 10497 | 12.23 |
| Guangxi | 2592 | 4728 | 6020 | 6961 | 18886 | 26862 | 34839 | 10342 | 10.95 |
| Anhui | 1446 | 1929 | 13168 | 8205 | 19846 | 29419 | 38991 | 10259 | 14.09 |
| Sichuan | 1543 | 1809 | 6574 | 7596 | 20521 | 27325 | 34128 | 9809 | 13.19 |
| Guizhou | 3876 | 4034 | 4363 | 5576 | 18558 | 25934 | 33310 | 9002 | 8.99 |
| Henan | 1108 | 1356 | 5836 | 6815 | 17094 | 26922 | 36749 | 8724 | 15.03 |
| Gansu | 1043 | 1165 | 5604 | 5962 | 18358 | 25061 | 31763 | 8336 | 14.64 |
| Xizang | 1699 | 2039 | 5123 | 6001 | 13265 | 25935 | 38606 | 8244 | 13.31 |
| Yunnan | 1197 | 1455 | 5069 | 5597 | 17209 | 23096 | 28983 | 7914 | 13.6 |

## 4.Adjustment index of human resource quality

### 4.1. Index system to measure human resource quality

Human resources are the sum of intelligence, physical strength and creativity condensed in the human body. Scholars generally study the quality level of human resources based on workers' years of education, health status, innovation ability, production efficiency and other aspects. In practice, according to the meaning and characteristics of human resources and the availability of data, the quality of human resources in China can be measured from four dimensions: education level, medical security, innovation ability and production efficiency. These 4 dimensions are taken as the Grade I index to measure the quality of human resources. The corresponding 10 Grade II indexes are shown in Table 2.

Table 2 shows that the "education level" corresponds to the 4th grade II index, denoted $X_1$-$X_4$. "Medical security" corresponds to two grade II indexes, $X_5$ and $X_6$. "Innovation capability" corresponds to three grade II indexes, denoted $X_7$-$X_9$. "Production efficiency" corresponds to indicator $X_{10}$.

Table 2. The Index system of HR quality.

| Grade I index | Grade II index | The index code |
|---|---|---|
| Education level | Years of schooling per person aged 6 and above (years) | $X_1$ |
| | Proportion of population with college degree or above (%) | $X_2$ |
| | Proportion of illiterate population aged 15 and above (%) | $X_3$ |
| | Per capita education expenditure (yuan/person) | $X_4$ |
| Medical security | Per capita expenditure on health (yuan/person) | $X_5$ |
| | Number of health technicians per 1,000 persons (person) | $X_6$ |
| Innovation ability | Number of patent applications per 10,000 people (number) | $X_7$ |
| | Number of patents granted per 10,000 persons (number) | $X_8$ |
| | Employees in research and development institutions as a percentage of employees (%) | $X_9$ |
| Production efficiency | Labour productivity (yuan/person) | $X_{10}$ |

## 4.2. Hedonic price method

**4.2.1 Principle of the hedonic price method.** According to the analysis of Section 2.5, this paper adopts the hedonic price method to make quality adjustments to human resource prices. The hedonic method currently has three forms: the linear form, the double log form, and the semilog form. Rosen [34], Silver [35] and Diewert [36] noted that the assumption of a logarithmic function is closer to reality. This is because the dimensions of the factors that affect the quality of human resources are different, and the value of the difference is large. Taking the logarithm of the variable on both sides of the double logarithm model can better eliminate the variation range of the variable. Therefore, this paper chooses the double logarithm model for hedonic price analysis.

The hedonic method for price adjustment must select the right characteristic index. The selected indicators can represent the quality characteristics of the commodity, and these quality characteristics should have a high correlation with the commodity price. The change in quality is deducted from the change in price to obtain the change in the "pure price" of the commodity.

The hedonic characteristic regression equation makes use of the characteristics and endowments of commodities to fully reflect the impact of commodity characteristics on prices and the changes in commodity prices in different periods. This method has the advantage of information completeness compared with other methods. The characteristic price index method must establish a hedonic regression model in the time period (usually one year) within the survey time range. The price regression of the initial year is regarded as the base period value, and the price characteristics of the remaining years are regarded as the reporting period value. According to the general price index method, the price index with different commodity characteristics in different periods can be obtained.

Next, taking the double logarithm model as an example, the hedonic regression equation of commodity price and commodity characteristics is established:

$$\ln P_k^t = \alpha_t + \sum_{n=1}^{N} \beta_{n,k} \ln X_{nk}^t + \varepsilon_k^t \tag{6}$$

$$\ln P_k^0 = \alpha_0 + \sum_{n=1}^{N} \beta_{n,k} \ln X_{nk}^0 + \varepsilon_k^0 \tag{7}$$

Eqs ([6]) and ([7]) are the characteristic regression equations of the reporting period and the base period, respectively. The regression coefficients of Eqs ([6]) and ([7]) are estimated by least squares regression, and the regression equations are obtained.

Fixed commodity characteristics in the base period, establishing the Laspeyres price index:

$$LHI_{t,0} = \frac{\exp(\alpha_t + \sum \beta_{n,t}\ln X_n^0)}{\exp(\alpha_0 + \sum \beta_{n,0}\ln X_n^0)} \quad (8)$$

The commodity characteristics were fixed in the reporting period, establishing the Paasche price index:

$$PHI_{t,0} = \frac{\exp(\alpha_t + \sum \beta_{n,t}\ln X_n^t)}{\exp(\alpha_0 + \sum \beta_{n,0}\ln X_n^t)} \quad (9)$$

In Formula ([6]) to Formula ([9]), $P_k^t$ is the price of human resources in region k in year t. $X_{nk}^t$ is the human resource quality characteristics of the n commodity in region k in year t. $\beta_{n,k}$ is the regression coefficient of human resource quality characteristics of the n commodity in region k. $t = 1,2,\cdots,T, k = 1,2,\cdots,N$. $\alpha$ is the constant term. $\varepsilon$ represents the error term. The superscript $t$ represents the reporting period and the superscript 0 represents the base period.

Because the Laspeyres index cannot reflect the change in quantity structure, the Paasche index cannot eliminate the influence of weight on relative change. To avoid these defects, this paper uses the Fischer index of Formula ([10]) to compile the "Human Resource Quality Adjustment Index" (Referred to as the *HI*). Here, the Fischer index is the square root of the Laspeyres index and the Paasche index.

$$HI_{t,0} = \sqrt{LHI_{t,0} \times PHI_{t,0}} \quad (10)$$

In Formulas ([8])–([10]), LHI, PHI and HI respectively represent Laspeyres price index, Paasche price index and Human Resource Quality Adjustment Index.

**4.2.2 Estimation procedure of the HR quality adjustment index.**

1. **Select Typical Attributes**
   Among the influencing factors of commodity prices, commodity quality is an important factor. According to the human resources quality index system, the quality of human resources is studied from four aspects: education level, medical security, innovation ability and production efficiency. The 10 Grade II indices $X_1$-$X_{10}$ in Table 2 are the quality attributes of human resources.

2. **Select the Appropriate Characteristic Price Function of Human Resources**
   Considering that the magnitude of some indicators in the human resource quality indicators $X_1$-$X_{10}$ is relatively large, the double logarithmic model is selected. We can measure the percentage change in the dependent variable for every 1% change in the independent variable.

3. **Principal Component Regression is Performed for Each Year Variable**
   First, principal component analysis is conducted on the selected attribute variable $X_i$ ($i = 1,2.\ldots.$). We select the principal component that can represent most of the information of the original variable, denoted as $prin_i$ ($i = 1,2.\ldots.$). Second, the regression analysis is conducted with human resource price $P_i$ as the dependent variable and principal component variable$prin_i$ as the independent variable. Then, taking the principal component variable $prin_i$ as the dependent variable and attribute variable $X_i$ as the independent variable,

regression analysis is conducted. Finally, the regression equation is transformed into the form of human resource price $P_i$ as the dependent variable and attribute variable $X_i$ as the independent variable.

4. **Compile the Human Resources Quality Adjustment Index**
   The human resource quality adjustment index ($HI$) is compiled according to Eq (6)–Eq (10).

## 4.3 Estimation process and results of HR quality adjustment index

**4.3.1 Estimation process.** The hedonic double log model is used to carry out cross-sectional data regression for 31 provinces (autonomous regions and municipalities) by year (1995–2020), with the logarithm of human resource price (ln $P$) as the dependent variable and the logarithm of human resource quality indices (ln $X_1$-ln $X_{10}$) as the independent variable. The hedonic double logarithm model is further improved, and the following results are obtained:

$$\ln P_k^t = \beta_0 + \sum_{i=1}^{10} \beta_{ki} \ln X_{ki}^t + \varepsilon_k^t \tag{11}$$

In Formula (11), $P_k^t$ is the human resource price in the $t$-year $k$-area and $X_{ki}^t$ is the $i$ kind of HR quality characteristic in the $t$-year $k$-area. $\beta_0$ is the intercept term. $\varepsilon_k^t$ is the random error term. Here, $t$ = 1995,1996,. . .,2020. $k$ represents the 31 provinces of China, $k$ = 1,2, . . ., 31. $i$ represents the human resource quality index, $i$ = 1,2, . . ., 10, which is the quality index $X_1$-$X_{10}$ in Table 2. The economic implication of Eq (11) is the effect of the changes in the quality of human resources on the price of human resources. It can be responded to by the magnitude of the coefficient $\beta$.

Due to the large number of independent variables, it is easy to produce multiple collinearity. First, the independent variables of each year are dimensionally reduced by principal component analysis, and then regression analysis is conducted with ln $P$ as the dependent variable and the principal component variables as the independent variables. In the following, the data of 1995 and 2020 are used as examples to conduct principal component regression on the price and quality of human resources in 31 provinces of China.

First, taking 1995 as an example, a principal component regression is performed. Principal component analysis is conducted on 10 independent variables in 1995, and the results are shown in Table 3. The statistical software SPSS is used here.

Table 3 shows the total variance table explained by the indicators of human resource quality in 31 provinces in 1995. The variance of the first 4 principal components accounted for

**Table 3. Explained total variance table of HR quality indicators in 31 provinces in 1995.**

| Compo-nent | Initial eigenvalue | | | Extract sum of squares and load | | | Rotate sum of squares load | | |
|---|---|---|---|---|---|---|---|---|---|
| | Sum | % of the variance | The cumulative % | Sum | % of the variance | The cumulative % | Sum | % of the variance | The cumulative % |
| 1 | 6.4915 | 64.92 | 64.92 | 6.49 | 64.92 | 64.92 | 3.78 | 37.84 | 37.84 |
| 2 | 1.6547 | 16.55 | 81.46 | 1.65 | 16.55 | 81.46 | 3.14 | 31.42 | 69.26 |
| 3 | 0.9149 | 9.15 | 90.61 | 0.91 | 9.15 | 90.61 | 1.32 | 13.20 | 82.47 |
| 4 | 0.3671 | 3.67 | 94.28 | 0.37 | 3.67 | 94.28 | 1.18 | 11.82 | 94.28 |
| 5 | 0.2927 | 2.93 | 97.21 | | | | | | |
| 6 | 0.1054 | 1.05 | 98.26 | | | | | | |
| 7 | 0.0721 | 0.72 | 98.98 | | | | | | |
| 8 | 0.0519 | 0.52 | 99.50 | | | | | | |
| 9 | 0.0382 | 0.38 | 99.89 | | | | | | |
| 10 | 0.0115 | 0.11 | 100.00 | | | | | | |

approximately 94.28% of the total variance, which retained most of the information of the original variables. In this way, the original 10 indicators reflecting the quality of human resources are transformed into 4 new indicators, recorded as $prin1$, $prin2$, $prin3$, $prin4$. Here, $\ln P$ is taken as the dependent variable and $prin1$, $prin2$, $prin3$, $prin4$ as the independent variables to conduct ordinary least squares regression, and the principal component regression equation is obtained as follows:

$$\ln P = 7.7560 + 0.0121 prin1 + 0.2300 prin2 + 0.0459 prin3 - 0.0401 prin4 \tag{12}$$

Regression Formula (12) passes the significance test, and the four principal components $prin1$, $prin2$, $prin3$, $prin4$ are significant at the 5% level. Next, $prin1$, $prin2$, $prin3$ and $prin4$ are successively taken as the dependent variables, and the 10 Grade II indexes $X_1$ - $X_{10}$ in Table 2 are taken as the independent variables to conduct the ordinary least squares regression estimation. The results are as follows:

$$prin1 = 4.5800 + 3.7698\ln X_1 - 0.2077\ln X_2 - 1.5142\ln X_3 - 0.3138\ln X_4 - 0.9312\ln X_5 \\ -0.6870\ln X_6 + 0.6970\ln X_7 + 0.6880\ln X_8 - 0.2242\ln X_9 + 0.6571\ln X_{10} \tag{13}$$

$$prin2 = -7.8044 - 1.3834\ln X_1 - 0.1125\ln X_2 + 0.4581\ln X_3 + 1.0874\ln X_4 + 1.0048\ln X_5 \\ +0.0044\ln X_6 + 0.1755\ln X_7 + 0.0598\ln X_8 - 0.2086\ln X_9 + 0.8803\ln X_{10} \tag{14}$$

$$prin3 = -1.1640 + 0.0917\ln X_1 + 1.0056\ln X_2 + 0.7664\ln X_3 - 0.0844\ln X_4 - 0.0971\ln X_5 \\ -0.4865\ln X_6 + 0.0924\ln X_7 + 0.1640\ln X_8 - 0.0134\ln X_9 - 0.1539\ln X_{10} \tag{15}$$

$$prin4 = 1.2766 + 0.1627\ln X_1 - 0.2417\ln X_2 - 0.5832\ln X_3 - 0.2212\ln X_4 + 0.0101\ln X_5 \\ +1.3299\ln X_6 - 0.2397\ln X_7 - 0.2298\ln X_8 + 0.5327\ln X_9 - 0.4719\ln X_{10} \tag{16}$$

The reduced principal component regression formula is

$$\ln P = 7.0314 + 0.1678\ln X_1 + 0.0315\ln X_2 - 0.0905\ln X_3 + 0.0942\ln X_4 + 0.0670\ln X_5 \\ +0.1389\ln X_6 + 0.0572\ln X_7 + 0.0478\ln X_8 + 0.0472\ln X_9 + 0.0941\ln X_{10} \tag{17}$$

Next, principal component regression is conducted on the human resource quality index in 2020. The principal component regression formula is

$$\ln P = 10.1704 + 0.0973 prin1 + 0.1389 prin2 + 0.0429 prin3 + 0.1348 prin4 \tag{18}$$

The reduced principal component regression formula is

$$\ln P = 7.5418 + 0.3821\ln X_1 + 0.1266\ln X_2 - 0.0540\ln X_3 + 0.0491\ln X_4 + 0.0088\ln X_5 \\ +0.2712\ln X_6 + 0.06482\ln X_7 + 0.0642\ln X_8 + 0.0484\ln X_9 + 0.1468\ln X_{10} \tag{19}$$

Similarly, principal component regression is conducted on the indicators of the remaining years, and the regression results of each year can be obtained. The regression coefficients of the past years passed the significance test under the principal component analysis. The $F$ value of the regression formula is greater than the critical value, and the regression formula is equally significant. Descriptive statistics of the regression coefficients for each regression formula in 1995–2020 are shown in Table 4.

Table 4 shows the mean, change interval and change amplitude of the regression coefficients of each variable.

**Table 4. Descriptive statistical results of each regression coefficient from 1995 to 2020.**

| Variable | Regression coefficient | Minimum value | Maximum value | Mean | Standard deviation |
|---|---|---|---|---|---|
| Constant | $\beta_0$ | 6.0368 | 7.5531 | 6.7949 | 0.4028 |
| Years of schooling per person aged 6 and above ($X_1$) | $\beta_1$ | 0.0431 | 0.5402 | 0.2694 | 0.1414 |
| Proportion of population with college degree or above ($X_2$) | $\beta_2$ | 0.0315 | 0.1703 | 0.0998 | 0.0436 |
| Proportion of illiterate population aged 15 and above ($X_3$) | $\beta_3$ | -0.0991 | -0.0106 | -0.0616 | 0.0252 |
| Per capita education expenditure ($X_4$) | $\beta_4$ | 0.0491 | 0.1940 | 0.1216 | 0.0368 |
| Per capita expenditure on health ($X_5$) | $\beta_5$ | 0.0088 | 0.1559 | 0.0771 | 0.0404 |
| Number of health technicians per 1,000 persons ($X_6$) | $\beta_6$ | 0.1268 | 0.3421 | 0.2315 | 0.0690 |
| Number of patent applications per 10,000 people ($X_7$) | $\beta_7$ | 0.0311 | 0.0909 | 0.0508 | 0.0116 |
| Number of patents granted per 10,000 persons ($X_8$) | $\beta_8$ | 0.0304 | 0.0995 | 0.0506 | 0.0133 |
| Employees in research and development institutions as a percentage of employees ($X_9$) | $\beta_9$ | 0.0396 | 0.0661 | 0.0564 | 0.0076 |
| Labour productivity ($X_{10}$) | $\beta_{10}$ | 0.0886 | 0.1505 | 0.1207 | 0.0232 |

First, except for the "proportion of illiterate population over 15 years old" ($X_3$), which has a negative impact on the price of human resources, all other variables have a positive impact.

Second, the factors affecting the price of human resources from strong to weak are "education level", "medical security", "production efficiency" and "innovation ability". The top two influencing coefficients on the price of human resources are "years of schooling per person aged 6 and above" ($X_1$) and "number of health technicians per 1,000 persons" ($X_6$), and their regression coefficient ranges are (0.0431, 0.5402), (0.1268, 0.3421), respectively. Next, in order of the influence coefficient from high to low, the index is "per capita education expenditure" ($X_4$), "labour productivity" (X10), "proportion of population with college degree or above" ($X_2$), "per capita expenditure on health" ($X_5$), "employees in research and development institutions as a percentage of employees" ($X_9$), "number of patent applications per 10,000 people" ($X_7$), and "number of patents granted per 10,000 persons" ($X_8$).

**4.3.2 Estimation results.** The human resource quality adjustment index (*HI*) of each province can be obtained by substituting the human resource prices of 31 provinces (autonomous regions and municipalities) in each year and the regression coefficients calculated according to Section 4.3.1 into Eqs (6)–(10). Fig 4 shows the average level of *HI* in 31 provinces (autonomous regions and municipalities) in China from 1995 to 2020 and ranks the index from high to low.

Fig 4 shows the following.

First, provinces with a high human resource quality adjustment index generally have a higher human resource quality level or more human resources. The top five provinces in *HI* are Chongqing, Beijing, Jiangsu, Shaanxi and Shanghai, and the means of the human resource quality adjustment index are 2.0006, 1.9995, 1.9836, 1.9726 and 1.9242, respectively. Six of the top 10 provinces are developed regions and provinces in the east, while the four noneastern provinces are Chongqing, Shanxi, Guizhou and Henan.

Second, provinces with low *HI* have relatively weak economic development levels and weak human resource quality. The bottom five provinces are Jilin, Liaoning, Yunnan, Hebei and Guangxi, with average values of the human resource quality adjustment index of 1.7369, 1.7340, 1.7137, 1.7077 and 1.6981, respectively. Most of the bottom 10 provinces are located in the central and western regions, with three western provinces (Xizang, Yunnan and Guangxi), three central provinces (Heilongjiang, Jiangxi and Jilin), and four eastern provinces (Hainan, Fujian, Liaoning and Hebei).

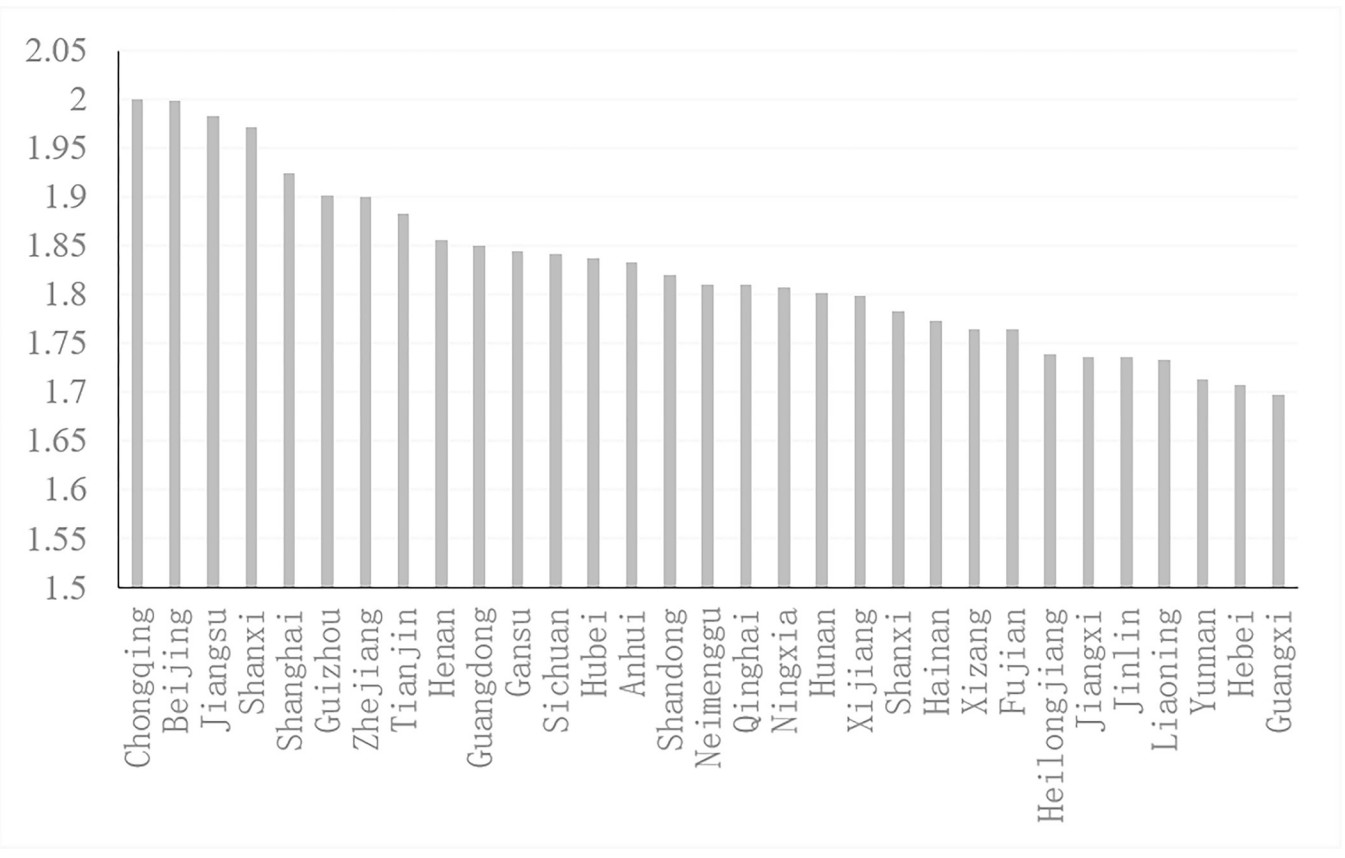

**Fig 4. Mean values of the quality adjustment index of HR in 31 provinces.**

## 5.Quality adjustment results of human resource prices

The human resource quality index system is introduced above, and the human resource quality adjustment index ($HI$) is calculated by using the hedonic characteristic price index method. The "HR price without quality adjustment" is called the "nominal human resource price" and denoted $P_1$. The "quality-adjusted HR price" is called the "actual human resources price" and denoted $P_0$. The relationship of the three parts is as follows: $P_0 = P_1/HI$.

After adjusting the quality of human resources, the nominal human resources price and the actual human resources price have a gap. This is because the nominal price of human resources includes the quality factor of human resources. After adjusting the quality of human resources, only the "pure" price of human resources is left, and the value will decrease accordingly.

### 5.1. Comparison of human resource prices in different provinces

Table 5 shows the mean and ratio of the nominal human resource price and actual human resource price sorted according to the ratio of the two human resource prices.

As seen in Table 5, first, the actual price of human resources decreases significantly, and the multiple of the price of human resources before and after quality adjustment is between 1.75 and 2.12.

Second, the provinces with a relatively large difference between the nominal and actual human resource prices are Jiangsu, Shaanxi, Chongqing, Beijing, and Henan, where the

**Table 5. Comparison of the average nominal and actual HR prices in 31 provinces.**

| Province | Average price of nominal human resources (yuan/person) | Average price of actual human resources (yuan/person) | Ratio of nominal to actual price means |
|---|---|---|---|
| Jiangsu | 18258 | 8606 | 2.12 |
| Shanxi | 11580 | 5472 | 2.12 |
| Chongqing | 13262 | 6341 | 2.09 |
| Beijing | 34490 | 16521 | 2.09 |
| Henan | 8724 | 4215 | 2.07 |
| Gansu | 8336 | 4044 | 2.06 |
| Tianjin | 21641 | 10695 | 2.02 |
| Qinghai | 10782 | 5337 | 2.02 |
| Guizhou | 9002 | 4476 | 2.01 |
| Ningxia | 12045 | 6023 | 2.00 |
| Sichuan | 9809 | 4915 | 2.00 |
| Anhui | 10259 | 5183 | 1.98 |
| Hunan | 11405 | 5779 | 1.97 |
| Shandong | 13420 | 6809 | 1.97 |
| Shanghai | 37518 | 19091 | 1.97 |
| Hubei | 10497 | 5354 | 1.96 |
| Zhejiang | 17715 | 9147 | 1.94 |
| Neimenggu | 13969 | 7216 | 1.94 |
| Xizang | 8244 | 4300 | 1.92 |
| Hainan | 11569 | 6055 | 1.91 |
| Shanxi | 13503 | 7071 | 1.91 |
| Xijiang | 12629 | 6675 | 1.89 |
| Yunnan | 7914 | 4256 | 1.86 |
| Jiangxi | 11807 | 6363 | 1.86 |
| Guangdong | 23403 | 12694 | 1.84 |
| Hebei | 12314 | 6689 | 1.84 |
| Liaoning | 14840 | 8091 | 1.83 |
| Heilongjiang | 11569 | 6319 | 1.83 |
| Jinlin | 12242 | 6782 | 1.81 |
| Fujian | 15596 | 8674 | 1.80 |
| Guangxi | 10342 | 5901 | 1.75 |

nominal human resource prices are 2.12 times, 2.12 times, 2.09 times, 2.09 times and 2.07 times the actual price, respectively. Liaoning, Heilongjiang, Jilin, Fujian and Guangxi have small differences in nominal and actual human resource prices; the differences are 1.83 times, 1.83 times, 1.81 times, 1.80 times and 1.75 times, respectively.

## 5.2. Ranking changes in nominal and actual human resource prices by province

Fig 5 shows the average level of nominal and actual human resource prices in 31 provinces of China from 1995 to 2020 and ranks the mean values of nominal human resource prices from highest to lowest.

Fig 5 shows the following.

First, the top five provinces in terms of the mean value of nominal human resource prices are all developed provinces in the east, while the underdeveloped provinces are mainly located in the western region. The top five provinces are Shanghai, Beijing, Guangdong, Tianjin and

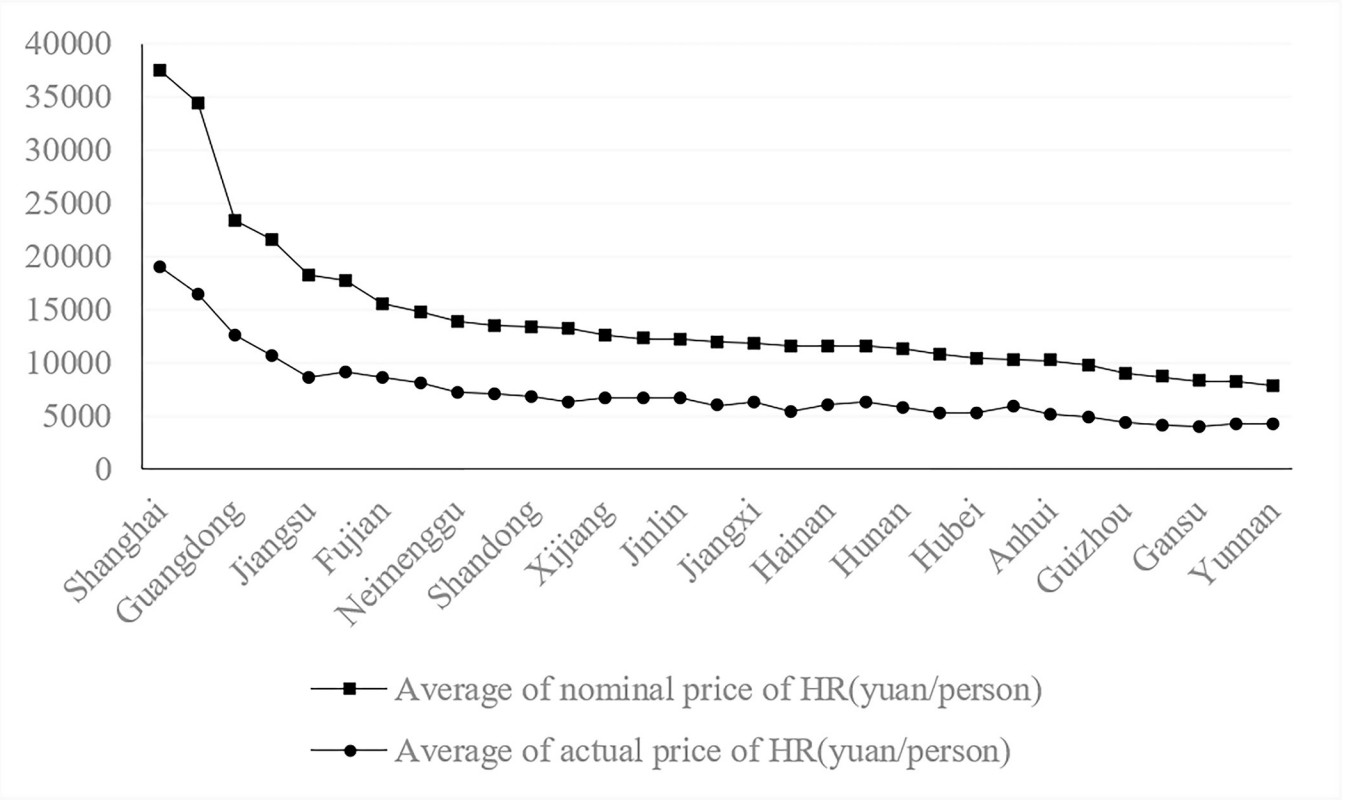

**Fig 5. Average nominal and actual HR prices of 31 provinces from 1995 to 2020.**

Jiangsu, and their average prices of human resources are 37,518 yuan/person, 34,490 yuan/person, 23,403 yuan/person, 21,641 yuan/person and 18,258 yuan/person, respectively. The bottom five provinces are Guizhou, Henan, Gansu, Xizang and Yunnan, with average human resource prices of 9,002 yuan/person, 8,724 yuan/person, 8,336 yuan/person, 8,244 yuan/person and 7,914 yuan/person, respectively.

Second, after quality adjustment, the top five provinces in terms of actual human resource prices are Shanghai, Beijing, Guangdong, Tianjin and Zhejiang, with average human resource prices of 19,091 yuan/person, 16,521 yuan/person, 12,694 yuan/person, 10,695 yuan/person and 9,147 yuan/person, respectively; these are all developed provinces in the east. The bottom five provinces are Guizhou, Xizang, Yunnan, Henan and Gansu, with average human resource prices of 4,476 yuan/person, 4,300 yuan/person, 4,256 yuan/person, 4,215 yuan/person and 4,044 yuan/person, respectively.

Third, compared with before and after quality adjustment, the ranking of human resource prices in each province does not change substantially. The provinces that increased significantly were Jilin, Jiangxi, Heilongjiang, and Guangxi. Chongqing, Ningxia and Shanxi are the provinces that dropped significantly. The HR prices remain in the top position. The more stable regions are Shanghai, Beijing, Guangdong, and Tianjin, while the less stable provinces are Guizhou, Henan, Gansu, Xizang and Yunnan.

Therefore, the provinces with the highest human resource prices consistently have higher human resource quality levels and higher "pure" human resource prices. The provinces whose human resource prices have been lagging indicate that both the quality level and the "pure"

price of human resources are lagging. These regions need to pay attention to improving the quality and the salary treatment of human resources.

## 6.Conclusions and recommendations

### Research conclusions

Compared with the previous studies, the findings of this paper include the development of the quantity of human resources in China. This paper compares the differences between the nominal and actual human resource price across regions. The changes in the quality of human resources are analyzed. The research conclusions of this paper are as follows:

1. The number of human resources in China shows a trend of rising first and then declining from 1995 to 2020, with an average growth rate of 0.39%. The decline began in 2016 (by 290,000 people), and the decline in the number of human resources in 2019 and 2020 was even greater (by 3.35 million and 3.83 million, respectively). Among the 31 provinces, Shandong, Henan and Guangdong ranked in the top three in terms of human resources, while Ningxia, Qinghai and Xizang had fewer human resources.

2. Nominal human resource prices in China rose from 1995 to 2020, fell slightly in 2013 and have since continued to rise. In 2020, the national nominal human resource price was 33,707 yuan/person, 15.44 times that of 1995. In terms of the average price level of human resources in each province, the price of human resources in Shanghai, Beijing, Guangdong, Tianjin and Jiangsu leads the way, while the price of human resources in Guizhou, Henan, Gansu, Yunnan and Xizang is lower.

3. Six of the top 10 provinces in the human resources quality adjustment index are located in developed eastern regions, while the four non-eastern provinces are Chongqing, Shanxi, Guizhou and Henan. The provinces with high human resource quality adjustment index generally have high quality human resource level, or have a large number of human resources. Provinces with low quality adjustment index correspond to lower level of economic development.

4. The price of human resources after quality adjustment has decreased significantly, and the multiple difference between the price of human resources and that of nominal human resources is between 1.75 and 2.12. The top five provinces in terms of quality-adjusted actual human resource prices are Shanghai, Beijing, Guangdong, Tianjin and Zhejiang, while the bottom five provinces are Guizhou, Yunnan, Henan, Xizang and Gansu.

5. After the adjustment of human resources quality, the price ranking of human resources in each province has little change. Jilin, Jiangxi, Heilongjiang and Guangxi made significant gains in the rankings. Chongqing, Ningxia and Shanxi are among the provinces that dropped significantly in the rankings. The price of human resources has always been in the top, and the ranking is relatively stable are Shanghai, Beijing, Guangdong, Tianjin. Consistently at the bottom of the list are Guizhou, Henan, Gansu, Xizang and Yunnan.

6. The quality of China's human resources has been rising and has been optimized in terms of "education level", "medical care", "innovation capacity" and "productivity". The quality of China's human resources has been continuously optimized in terms of "education level", "medical protection", "innovation capacity" and "productivity". The average year of education in China rose from 6.74 years in 1995 to 9.9 years in 2020. The number of hospital beds per 10,000 people rose from 25.93 in 1995 to 64.6 in 2020. The proportion of R&D expenditure in GDP rose from 0.50% in 1995 to 2.4% in 2020. The labor productivity rose

from 0.9 million yuan per person in 1995 to 12.7 million yuan per person in 2020. The life expectancy rose from 70.80 years in 1995 to 77.3 years in 2020. The average age of workers rose from 34.68 years in 1995 to 39 years in 2020.

## 6.2. Recommendations

This paper studied the human resource price of China based on the macro perspective, used the hedonic price model method to adjust the human resource price, and compared the difference in the human resource price after quality adjustment. Based on this, this paper presents the following suggestions.

1. Improving the data collection mechanism that are related to the accounting for the value of human resources. The country should pay attention to the impact and contribution of human resources to economic development. According to the World Bank's theory of national wealth accounting, the contribution of human resources to economic growth is becoming increasingly important. In order to accurately account for the value of China's human resources requires complete data support, and the data that need to be improved include the employment rate, unemployment rate, health status and labor income of the working population by age group, and the number of people entering and exiting the labor market in previous years. The human capital needs to be incorporated into the national economic accounting system. The opening stock, current change, depreciation and closing stock of human capital need to be explicitly included in the framework of the accounting system. Providing a theoretical and practical basis and guarantee for China's human resource value accounting.

2. Promoting the rational allocation of human resources among regions. The central and western regions need to build a high-level open economy, strengthen interprovincial economic cooperation and exchanges, and attract talent to migrate to the central and western regions. It is necessary to open up the manufacturing sector in an orderly manner, gradually ease the entry threshold for the service sector, and encourage all types of talent to fully play their effective roles. Education level has a significant impact on the quality of regional human resources, and the investment of regional educational resources cannot be relaxed. The introduction of human resources should be strengthened to attract and provide adequate living security for high-quality talent. The rational allocation of human resources among regions is an important guarantee to promote the balanced development of the economy.

3. Raising wages for human resources in the central region. Society has changed from seizing natural resources to seizing human resources. Raising the salary level and attracting talent is an important measure to implement the strategy of the "rise of central China". The price of human resources in the central region is low, lagging behind the eastern and western regions. Government departments in the central region should adapt measures to local conditions and carry out activities to attract investment in line with local wage levels to enhance local innovation capacity and economic vitality. While cultivating high-quality talent, attention should be given to improving the salary and treatment of grassroots staff and ensuring the enthusiasm of grassroots staff to provide comprehensive and effective economic security for society.

4. Improving the quality of human resources in the western region. The quality of human resources in the western region generally lags behind that in the eastern and central regions. The western region needs to improve its level in terms of higher education, innovation

ability and labour productivity and increase vocational skills education to train qualified graduates who can meet social needs. Furthermore, internship training should be increased so that graduates can understand the environment and needs of companies in advance. Furthermore, we suggest increasing the enrolment rate of senior high school students, strengthening the development of colleges and universities, and supporting the construction of "double first-class" colleges and universities in the western region in addition to actively carrying out vocational skills education and improving the quality of training and the vast number of workers in the western region so that the dividend model of economic growth can rely on the quality of the labour force. Funds can be concentrated to establish feasible industrial demonstration zones and science and technology transformation zones in the west. We can introduce advanced science and technology and excellent management concepts from the eastern region to the western region to apply them rationally and adapt measures to local conditions. China's working-age population is shrinking, and the fruitful results of the "demographic dividend" are gradually declining. While increasing the quantity of human resource development, more attention should be given to improving the quality to change the "demographic dividend" into the "human resources dividend". The western region needs to pay special attention to the introduction of human resources and the cultivation of human resources. The development level of human resources in western China is backward relatively. The "14th Five-Year Plan" and Vision 2035 point out that we should insist on the strategy of innovation-driven development and implement the strategy of strengthening the country with talents. In order to achieve the goal of strengthening the country with talents, the western region has to reserve the qualified and abundant human resources for economic construction.

### 6.3 Limitations and future research

The limitation of the research in this paper is the indicators selection of the human resources quantity. This paper uses the number of employees to represent the number of human resources and uses the effective number of human resources. Human resources include a more wide range, but the data are not easy to collect. With the further improvement of human resources accounting in the future, the use of indicators that could more comprehensively reflect the number of human resources. Thus, it could make the accounting of human resources more accurate.

The possible future research of this paper is the accounting of human resource value. With the improvement of human resource data, the present value of earnings method could be used to calculate the value of regional human resources. The key issue is to calculate the discount rate of human resource value.

## Supporting information

**S1 File.**
(DOCX)

## Author Contributions

**Data curation:** Lan Ma.

**Funding acquisition:** Lan Ma.

**Methodology:** Lan Ma.

**Project administration:** Lan Ma.

**Resources:** Lan Ma.

**Software:** Lan Ma.

**Supervision:** Lan Ma.

**Writing – original draft:** Lan Ma.

**Writing – review & editing:** Lan Ma.

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
