## [Decision Letter · Decision Letter 0]

19 Jan 2023

PONE-D-22-35578Quality Adjustment and Analysis of Human Resource Prices in China: Based on a Hedonic Price ModelPLOS ONE

Dear Dr. Ma,

Thank you for submitting your manuscript to PLOS ONE. After careful consideration, we feel that it has merit but does not fully meet PLOS ONE’s publication criteria as it currently stands. Therefore, we invite you to submit a revised version of the manuscript that addresses the points raised during the review process.

We look forward to receiving your revised manuscript.

Kind regards,

Uzair Aslam Bhatti

Academic Editor

PLOS ONE

“This work was supported by the Humanities and Social Science Research Project of Anhui Universities: Research on the mechanism and path of digital economy driving manufacturing transformation and upgrading in Anhui Province from the perspective of complex network (Project Number: SK2021A0696); the Humanities and Social Science Research Project of Anhui Universities: Research on data-driven innovation value chain and high Quality development of technology-intensive industries in Anhui Province (Project Number: SK2021A0694); a Doctoral Research Foundation Project at Suzhou University 2021: Research on the mechanism and path of Digital economy driving the transformation and upgrading of manufacturing industry in Anhui Province (Project Number: 2021BSK015); and the Online course of Anhui Province—National Economic Accounting(Project Number: 2021xskc097).”

“Conceptualization: Lan Ma.

Data curation: Qi Wang, Lan Ma.

Funding acquisition: Dalei Wang.

Methodology: Lan Ma, Qi Wang.

Project administration: Dalei Wang.

Resources: Lan Ma.

Software: Lan Ma, Qi Wang, Dalei Wang.

Supervision: Dalei Wang.

Writing – original draft: Lan Ma.

Writing – review & editing: Lan Ma, Qi Wang.”

Reviewers' comments:

Reviewer's Responses to Questions

**Comments to the Author**

1. Is the manuscript technically sound, and do the data support the conclusions?

Reviewer #1: Partly

Reviewer #2: Yes

2. Has the statistical analysis been performed appropriately and rigorously? 

Reviewer #1: Yes

Reviewer #2: Yes

3. Have the authors made all data underlying the findings in their manuscript fully available?

Reviewer #1: Yes

Reviewer #2: Yes

4. Is the manuscript presented in an intelligible fashion and written in standard English?

Reviewer #1: Yes

Reviewer #2: No

5. Review Comments to the Author

Reviewer #1: Congratulations to the author for conducting this research and submitting a paper to this journal. It is, however, necessary to address the following issues before resubmitting.

>It is apparent that the quality of the paper is not in accordance with the standards of this journal. Describe the challenges and novelty of this research in the abstract and introduction.

>There is no indication that the novelty of the paper has been validated by the author. The author should provide a clear description of their work.

>The related work does not support the problem identified in this study.

>This paper does not cite all the statistics that were used in the study.

>As far as the use of mathematical symbols is concerned, there are no adequate explanations in the text.

>There is no adequate mathematical explanation to support the claim.

>Add a literature review section. The author is advised to include recent research that addresses a similar issue. A summary table of existing studies with strengths and weaknesses should also be presented by the author. The literature from 2021 and 2022 should be included along with more recent studies in the paper to strengthen the literature review section.

>To interpret the results of the analysis, the author should consider the following papers:

https://doi.org/10.3390/atmos12101338

https://ieeexplore.ieee.org/document/9411587

https://ieeexplore.ieee.org/document/9628315

https://doi.org/10.15244/pjoes/142146

https://doi.org/10.3389/fenvs.2022.945628

https://doi.org/10.3390/su13084312

http://dx.doi.org/10.15244/pjoes/148065

Reviewer #2: The article can be improved further on following suggestions:

1 The communication of this article is weak, a throughout proofreading is needed.

2 Add contribution of this article in the end of introduction section.

3 Use proper statistical software for graphs.

6. PLOS authors have the option to publish the peer review history of their article (what does this mean?). If published, this will include your full peer review and any attached files.

Reviewer #1: No

Reviewer #2: No

---

## [Author Response · Author response to Decision Letter 0]

17 Mar 2023

Response to Reviewers

Reviewer #1：

1.Describe the challenges and novelty of this research in the abstract and introduction.

Reply:

New challenges and novelty in the abstract:

Firstly, with the Marx's labor value theory and Adam Smith's theory as the theoretical basis, the System of National Economic Accounting 2008 (SNA2008) and the System of Social and Demographic Statistics (SSDS) as the accounting criteria, the price level of human resources in China is calculated. Secondly, establishing the human resource quality index system to measure the human resource quality adjustment index. Third, the Hedonic method was used to adjust the human resource price of 31 provinces (autonomous regions and municipalities directly under the Central Government) in China, and the "pure price" of human resources was obtained without quality factors. Lastly, comparing the price of human resources before and after quality adjustment.

New challenges and novelty in the introduction:

The innovations of this paper are in following: First, constructing the human resources quality index system. Second, measuring the human resources quality adjustment index. Third, estimating the "pure price" of human resources after quality adjustment. This paper focuses on China's human resources and calculates the price of human resources in China from 1995 to 2020. 

2.There is no indication that the novelty of the paper has been validated by the author.

Reply:

Novelty1: First, constructing the human resources quality index system.

Verification in the paper: In Section 4.1, the Index System to Measure Human Resource Quality which is showed in Table 2.

Novelty2: Second, measuring the human resources quality adjustment index. 

Verification of the paper: In part 4.2.1 and Part 4.2.2, there are respectively "Principle of the Hedonic Price Method" and "Estimation Procedure of the HR Quality Adjustment Index".

Novelty3: Third, estimating the "pure price" of human resources after quality adjustment.

Verification of the paper: In part 5 of the essay, Comparison of Human Resource Prices in Different Provinces, Ranking Changes in Nominal and Actual Human Resource Prices by Province

3.This paper does not cite all the statistics that were used in the study.

Reply:

All statistics cited have been used. They are shown in Table 1- Table 5 and Figure 1- Figure 5.

The data used in the table are HR prices in 31 provinces of China (part of years), the average nominal and actual HR prices in 31 provinces. 

The data used in the figure are Amount of HR in China from 1995 to 2020, Average amount of HR and average growth rates in China's 31 provinces, National and suburban and rural HR prices from 1995 to 2020, Mean values of the quality adjustment index of HR in 31 provinces, Average nominal and actual HR prices of 31 provinces from 1995 to 2020.

4.As far as the use of mathematical symbols is concerned, there are no adequate explanations in the text.

Reply:

I have explained the mathematical notation adequately. Table 2 shows the code of each index in the human resource quality index system. In Part 4.2.1, the symbolic description of formula (6) - (10) is added.

5.There is no adequate mathematical explanation to support the claim.

Reply:

Formula (6) - (10) is the mathematical explanation of the human resource quality adjustment index, and formula (15) - (22) is the specific calculation process of the human resource quality adjustment index.

6.Add a literature review section. The author is advised to include recent research that addresses a similar issue. A summary table of existing studies with strengths and weaknesses should also be presented by the author. The literature from 2021 and 2022 should be included along with more recent studies in the paper to strengthen the literature review section.

Reply:

The latest literature review is added, such as those from 2021, 2022 and 2023. In part 2.1, 2.2, 2.3, 2.4 and 2.5, respectively.

The advantages and disadvantages of the existing literature are also summarized. At the end of Part 2.5, the content is as follows:

The existing literature provides sufficient theoretical basis for the study of human resource price, which can be measured by the wage level of labor force. However, the existing research on the quality of human resources is limited. Firstly, the construction of human resource quality index system is not comprehensive. Secondly, there is no research on the quality adjustment of human resource price.

6.To interpret the results of the analysis, the author should consider the following papers.

Reply:

I have carefully read the paper for reference. Conclusions of analysis are further explained in 6.1.Research Conclusions.

Reviewer #2:

1.The communication of this article is weak, a throughout proofreading is needed.

Reply:

The article has been thoroughly proofread for the language.

2.Add contribution of this article in the end of introduction section.

Reply:

The article has added some innovative points at the end of the introduction. The contents are as follows:

The innovations of this paper are in following: First, constructing the human resources quality index system. Second, measuring the human resources quality adjustment index. Third, estimating the "pure price" of human resources after quality adjustment. This paper focuses on China's human resources and calculates the price of human resources in China from 1995 to 2020. 

3.Use proper statistical software for graphs.

Reply:

Statistical software SPSS is used in this paper, which is explained in the upper text of Table 3.

---

## [Decision Letter · Decision Letter 1]

29 Aug 2023

PONE-D-22-35578R1Quality Adjustment and Analysis of Human Resource Prices in China: Based on a Hedonic Price ModelPLOS ONE

Dear Dr. Ma,

Thank you for submitting your manuscript to PLOS ONE. After careful consideration, we feel that it has merit but does not fully meet PLOS ONE’s publication criteria as it currently stands. Therefore, we invite you to submit a revised version of the manuscript that addresses the points raised during the review process.

We look forward to receiving your revised manuscript.

Kind regards,

Xingwei Li, Ph.D.

Academic Editor

PLOS ONE

Journal Requirements:

Additional Editor Comments:

After peer review, the deficiencies of this manuscript still require further improvement. Please revise the manuscript carefully in the light of the reviewers' comments and respond point by point.

Reviewers' comments:

Reviewer's Responses to Questions

**Comments to the Author**

1. If the authors have adequately addressed your comments raised in a previous round of review and you feel that this manuscript is now acceptable for publication, you may indicate that here to bypass the “Comments to the Author” section, enter your conflict of interest statement in the “Confidential to Editor” section, and submit your "Accept" recommendation.

Reviewer #2: All comments have been addressed

Reviewer #3: (No Response)

2. Is the manuscript technically sound, and do the data support the conclusions?

Reviewer #2: Yes

Reviewer #3: Partly

3. Has the statistical analysis been performed appropriately and rigorously? 

Reviewer #2: Yes

Reviewer #3: Yes

4. Have the authors made all data underlying the findings in their manuscript fully available?

Reviewer #2: Yes

Reviewer #3: Yes

5. Is the manuscript presented in an intelligible fashion and written in standard English?

Reviewer #2: Yes

Reviewer #3: Yes

6. Review Comments to the Author

Reviewer #2: The minor issues have been addressed in the article in a good manner and hence recommended for publication.

Reviewer #3: Report on: Quality Adjustment and Analysis of Human Resource Prices in China: Based on a Hedonic Price Model

Thank you for entrusting me with the opportunity to review the paper. I would like to commend you on your work and the interesting findings presented in the paper. However, I would like to provide some feedback regarding the introduction section.

After carefully examining the abstract section, I would like to provide some suggestions for improvement. It appears that the abstract contains lengthy sentences and lacks proper structuring. To enhance clarity and readability, it is recommended to break down long sentences into shorter, more concise ones. This will allow readers to grasp the main points more easily. Additionally, organizing the information in a logical manner will aid in understanding the key elements of the paper.

The Introduction is long and loosely connected. It should be improved and better written. The introduction should, very clearly, highlight the aims, method, results and contributions of the paper. Some of the elements of the conclusion could be used in this writing-up of the introduction.

One notable observation is the absence of citations to support the background and context of the study. The introduction serves as a foundation for the research by providing an overview of the existing literature in the field. By referring to relevant research articles, you can establish a framework for your study and highlight the gap that your research intends to address. This will also demonstrate your familiarity with the existing literature and contribute to the credibility of your work.

I understand that your research may be exploring new avenues or addressing unique aspects within the field. However, it is crucial to demonstrate that your work has been situated within the broader scholarly conversation. This will not only lend support to your study but also allow researchers in the field to more easily contextualize and engage with your research.

Additionally, I recommend consulting established journals in your field, as they often publish comprehensive literature reviews and cutting-edge research that may be suitable for citation.

I commend your efforts and the contribution this article can make to the research community. By addressing this concern and enhancing the introduction section with appropriate citations, your article will undoubtedly be strengthened and have a more significant impact on the field.

The discussion section is weak. Kindly improve it. Compare your findings with prior studies’ findings. There is a lack of economic discussion for the regression results of the threshold model.

There are many typo errors in this manuscript, especially for tables. Kindly cross check.

The policy suggestion section need improvement. It is weak in its present form.

The limitations and possible future research directions should be interpreted at the end of the paper, so that the future researchers who are interested in this important topic could carry out follow-up studies.

Please check the grammar of the full text and ask a native English speaker to double-check the edited paper.

7. PLOS authors have the option to publish the peer review history of their article (what does this mean?). If published, this will include your full peer review and any attached files.

Reviewer #2: **Yes: **Muhammad Bilal

Reviewer #3: No

---

## [Author Response · Author response to Decision Letter 1]

20 Oct 2023

Response to Reviewers

Reviewer #3：

1.After carefully examining the abstract section, I would like to provide some suggestions for improvement. It appears that the abstract contains lengthy sentences and lacks proper structuring. To enhance clarity and readability, it is recommended to break down long sentences into shorter, more concise ones. This will allow readers to grasp the main points more easily. Additionally, organizing the information in a logical manner will aid in understanding the key elements of the paper..

Reply:

I have reorganized the content of the abstract. The long sentences in the abstract are broken down into shorter ones. I also used a logical way to organize the paper.

The new summary is as follows:

The competition in the world has shifted from natural resources and capital resources to human resources. The human resources have become the real power source of the economic growth. Firstly, the price of human resources in China is calculated. Secondly, in order to measure the human resources quality adjustment index, the indicators system is constructed. Third, the Hedonic method is used to calculate the human resources "pure price" of 31 provinces (autonomous regions and municipalities directly under the Central Government) in China. The "pure price" has no the factor of human resources quality. Lastly, comparing the price of human resources before and after quality adjustment. The results show that: first, the number of human resources in China increased continuously during 1995-2015 and decreased during 2016-2020. Second, the price of nominal human resources in China keeps rising. In 2020, the nominal price is 39,087 yuan per person which is 15.44 times as many as in 1995. Thirdly, after the quality adjustment, the price of human resources has decreased significantly. The multiple between the actual and nominal price of human resources is between 1.75 and 2.12. Fourthly, the province with high human resource quality adjustment index generally have high quality human resource level or quantity. Fifth, the top five provinces of actual human resource prices are Shanghai, Beijing, Guangdong, Tianjin, Zhejiang, the bottom five provinces are Guizhou, Yunnan, Henan, Xizang, Gansu. Finally, the paper puts forward some policy recommendations: Improving the data collection mechanism of human resources accounting to provide a basic guarantee for the accurate accounting of human resources. Improving the price of human resources in the central and western regions to attract the talents to transfer to the central and western regions. Enhancing the skills training of human resources to improve the quality of human resources in the western region.

2.The Introduction is long and loosely connected. It should be improved and better written. The introduction should, very clearly, highlight the aims, method, results and contributions of the paper. Some of the elements of the conclusion could be used in this writing-up of the introduction. One notable observation is the absence of citations to support the background and context of the study. The introduction serves as a foundation for the research by providing an overview of the existing literature in the field. By referring to relevant research articles, you can establish a framework for your study and highlight the gap that your research intends to address. This will also demonstrate your familiarity with the existing literature and contribute to the credibility of your work.

Reply:

(1)At the end of the introduction, the purpose, method and contribution of the paper are clarified. 

The aims of the research in this paper is as follows: First, exploring the status of the quantity and price of human resources in China. Second, establishing the human resource quality index system. Using the hedonic method to calculate the human resource quality adjustment index. Calculating the actual human resource prices. Finally, comparing and analyzing the difference between nominal and actual human resource prices. 

The research methods in this paper include the descriptive statistical analysis, the Hedonic feature price method, and the principal component analysis.

The contribution of this paper lies in the following: (1) Making up the gap in human resource price adjustment.It is conducive to improving the analytical framework of human resource price adjustment and quality analysis. (2) It is helpful for accelerating the improvement of the statistical practice of human resource accounting and expanding the content of population accounting. (3) This paper provides the basis for the government to formulate a reasonable policy of regional human resource allocation, which is an important impetus for the promotion of high-quality development of the economy and the implementation of the strategy of "Talents Strengthening the Nation".

(2) The research results of the paper are reflected in the last part of the paper, so it is not repeated in the introduction.

(3)The background and context of the study are added in the first paragraph of the introduction.

According to the "China Human Development Report 2019" released by the United Nations Development Program, China is one of the most rapidly advancing countries in the field of human development. Based on the Human Development Index (HDI), China has become a "high-level human development country" and is one of the fastest-growing countries in the field of human development in the past 30 years. China's Fourteenth Five-Year Plan puts forward the strategy of "Strengthening the Nation with Talents". In order to raise the level of human resources in the country, the cultivation, introduction, use and motivation of talents should be emphasized. Human resources are put into the labor market as labor that can create social wealth and value. 

(4)The related literature in this field are added in the introduction.

(5) The gap I should to address is the content of my contribution in the introduction. Making up the gap in human resource price adjustment.It is conducive to improving the analytical framework of human resource price adjustment and quality analysis. It is helpful for accelerating the improvement of the statistical practice of human resource accounting and expanding the content of population accounting. This paper provides the basis for the government to formulate a reasonable policy of regional human resource allocation, which is an important impetus for the promotion of high-quality development of the economy and the implementation of the strategy of "Talents Strengthening the Nation".

3.I understand that your research may be exploring new avenues or addressing unique aspects within the field. However, it is crucial to demonstrate that your work has been situated within the broader scholarly conversation. This will not only lend support to your study but also allow researchers in the field to more easily contextualize and engage with your research. Additionally, I recommend consulting established journals in your field, as they often publish comprehensive literature reviews and cutting-edge research that may be suitable for citation.

Reply: 

In the introduction and literature review, I add and collate the existing research. On the basis of summarizing the existing researches, the research content of this paper is put forward. I could put my work into a broader conversation.

The references I cited are all well-known journals.

4.The discussion section is weak. Kindly improve it. Compare your findings with prior studies’ findings. 

Reply: 

The discussion section has been added. I have compared the results of my study with the previous results.

Compared with the previous studies, the findings of this paper include the development of the quantity of human resources in China. This paper compares the differences between the nominal and actual human resource price across regions. The changes in the quality of human resources are analyzed. 

(6) The quality of China's human resources has been rising and has been optimized in terms of "education level", "medical care", "innovation capacity" and "productivity". The quality of China's human resources has been continuously optimized in terms of "education level", "medical protection", "innovation capacity" and "productivity". The average year of education in China rose from 6.74 years in 1995 to 9.9 years in 2020. The number of hospital beds per 10,000 people rose from 25.93 in 1995 to 64.6 in 2020. The proportion of R&D expenditure in GDP rose from 0.50% in 1995 to 2.4% in 2020. The labor productivity rose from 0.9 million yuan per person in 1995 to 12.7 million yuan per person in 2020. The life expectancy rose from 70.80 years in 1995 to 77.3 years in 2020. The average age of workers rose from 34.68 years in 1995 to 39 years in 2020.

5.There is a lack of economic discussion for the regression results of the threshold model.

Reply:

The economic implication of equation (11) is the effect of the changes in the quality of human resources on the price of human resources. It can be responded to by the magnitude of the coefficient .

The model (11) in this paper is the Hedonic model, not the threshold model. It reflects the relationship between human resource quality and human resource price. The economic implication of equation (11) is the effect of the changes in the quality of human resources on the price of human resources. It can be responded to by the magnitude of the coefficient .

6.There are many typo errors in this manuscript, especially for tables. Kindly cross check.

Reply:

I have corrected the mistakes in the text, especially those in the table carefully.

7.The policy suggestion section need improvement. It is weak in its present form.

Reply:

I have revised and improved the first and fourth points of the policy proposal.

(1)Improving the data collection mechanism that are related to the accounting for the value of human resources. The country should pay attention to the impact and contribution of human resources to economic development. According to the World Bank's theory of national wealth accounting, the contribution of human resources to economic growth is becoming increasingly important. In order to accurately account for the value of China's human resources requires complete data support, and the data that need to be improved include the employment rate, unemployment rate, health status and labor income of the working population by age group, and the number of people entering and exiting the labor market in previous years. The human capital needs to be incorporated into the national economic accounting system. The opening stock, current change, depreciation and closing stock of human capital need to be explicitly included in the framework of the accounting system. Providing a theoretical and practical basis and guarantee for China's human resource value accounting.

(4) Improving the quality of human resources in the western region. The quality of human resources in the western region generally lags behind that in the eastern and central regions. The western region needs to improve its level in terms of higher education, innovation ability and labour productivity and increase vocational skills education to train qualified graduates who can meet social needs. Furthermore, internship training should be increased so that graduates can understand the environment and needs of companies in advance. Furthermore, we suggest increasing the enrolment rate of senior high school students, strengthening the development of colleges and universities, and supporting the construction of "double first-class" colleges and universities in the western region in addition to actively carrying out vocational skills education and improving the quality of training and the vast number of workers in the western region so that the dividend model of economic growth can rely on the quality of the labour force. Funds can be concentrated to establish feasible industrial demonstration zones and science and technology transformation zones in the west. We can introduce advanced science and technology and excellent management concepts from the eastern region to the western region to apply them rationally and adapt measures to local conditions. China's working-age population is shrinking, and the fruitful results of the "demographic dividend" are gradually declining. While increasing the quantity of human resource development, more attention should be given to improving the quality to change the "demographic dividend" into the "human resources dividend". The western region needs to pay special attention to the introduction of human resources and the cultivation of human resources. The development level of human resources in western China is backward relatively. The "14th Five-Year Plan" and Vision 2035 point out that we should insist on the strategy of innovation-driven development and implement the strategy of strengthening the country with talents. In order to achieve the goal of strengthening the country with talents, the western region has to reserve the qualified and abundant human resources for economic construction.

8.The limitations and possible future research directions should be interpreted at the end of the paper, so that the future researchers who are interested in this important topic could carry out follow-up studies.

Reply:

At the end of the article, I added the limitations of the research and the possible direction of future research.

The limitation of the research in this paper is the indicators selection of the human resources quantity. This paper uses the number of employees to represent the number of human resources and uses the effective number of human resources. Human resources include a more wide range, but the data are not easy to collect. With the further improvement of human resources accounting in the future, the use of indicators that could more comprehensively reflect the number of human resources. Thus, it could make the accounting of human resources more accurate.

The possible future research of this paper is the accounting of human resource value. With the improvement of human resource data, the present value of earnings method could be used to calculate the value of regional human resources. The key issue is to calculate the discount rate of human resource value.

9.Please check the grammar of the full text and ask a native English speaker to double-check the edited paper.

Reply: 

I have checked the grammar in the paper and asked a native English speaker to double-check the editing.

Journal Requirements:

Reply: 

The paper does not cite references that have been retracted.

Some references are added and other Some references removed to increase the readability of the paper. It have been noted in the document of 'Revised Manuscript with Track Changes'.

---

## [Decision Letter · Decision Letter 2]

3 Jan 2024

Quality Adjustment and Analysis of Human Resource Prices in China: Based on a Hedonic Price Model

PONE-D-22-35578R2

Dear Dr. Ma,

We’re pleased to inform you that your manuscript has been judged scientifically suitable for publication and will be formally accepted for publication once it meets all outstanding technical requirements.

Kind regards,

Xingwei Li, Ph.D.

Academic Editor

PLOS ONE

Additional Editor Comments (optional):

Accept as is.

Reviewers' comments:

Reviewer's Responses to Questions

**Comments to the Author**

1. If the authors have adequately addressed your comments raised in a previous round of review and you feel that this manuscript is now acceptable for publication, you may indicate that here to bypass the “Comments to the Author” section, enter your conflict of interest statement in the “Confidential to Editor” section, and submit your "Accept" recommendation.

Reviewer #4: All comments have been addressed

2. Is the manuscript technically sound, and do the data support the conclusions?

Reviewer #4: Yes

3. Has the statistical analysis been performed appropriately and rigorously? 

Reviewer #4: Yes

4. Have the authors made all data underlying the findings in their manuscript fully available?

Reviewer #4: Yes

5. Is the manuscript presented in an intelligible fashion and written in standard English?

Reviewer #4: Yes

6. Review Comments to the Author

Reviewer #4: The authors had revised the paper according to four reviewer’s advices and suggestions. All the concerns from third reviewer had been addressed. The quality of the paper is improved. After two rounds of revision, the paper is valued to be documented.

One minor suggestion is that the unit should be added in the Y axis, for example figure 1 and figure 2. 6.3 can be added in front of section “Limitations and Future Research”

7. PLOS authors have the option to publish the peer review history of their article (what does this mean?). If published, this will include your full peer review and any attached files.

Reviewer #4: **Yes: **Min Ye

---

## [Editor Report · Acceptance letter]

24 Mar 2024

PONE-D-22-35578R2 

PLOS ONE

Dear Dr. Ma, 

I'm pleased to inform you that your manuscript has been deemed suitable for publication in PLOS ONE. Congratulations! Your manuscript is now being handed over to our production team.

Kind regards, 

on behalf of

Prof. Dr. Xingwei Li 

Academic Editor

PLOS ONE